# A Novel Channel Estimation Framework in MIMO Using Serial Cascaded Multiscale Autoencoder and Attention LSTM with Hybrid Heuristic Algorithm

**DOI:** 10.3390/s23229154

**Published:** 2023-11-13

**Authors:** B. M. R. Manasa, Venugopal Pakala, Ravikumar Chinthaginjala, Manel Ayadi, Monia Hamdi, Amel Ksibi

**Affiliations:** 1School of Electronics Engineering, Vellore Institute of Technology, Vellore 632014, India; manasaece55@gmail.com (B.M.R.M.); venugopalp.eee@gmail.com (V.P.); cvrkvit@gmail.com (R.C.); 2Department of Information Systems, College of Computer and Information Sciences, Princess Nourah bint Abdulrahman University, P.O. Box 84428, Riyadh 11671, Saudi Arabia; amelksibi@pnu.edu.sa; 3Department of Information Technology, College of Computer and Information Sciences, Princess Nourah bint Abdulrahman University, P.O. Box 84428, Riyadh 11671, Saudi Arabia; mshamdi@pnu.edu.sa

**Keywords:** channel estimation scheme, multiple input multiple output channel, hybrid serial cascaded network, revised position-based wild horse and energy valley optimizer, long short term memory, autoencoder

## Abstract

In wireless communication, multiple signals are utilized to receive and send information in the form of signals simultaneously. These signals consume little power and are usually inexpensive, with a high data rate during data transmission. An Multi Input Multi Output (MIMO) system uses numerous antennas to enhance the functionality of the system. Moreover, system intricacy and power utilization are difficult and highly complicated tasks to achieve in an Analog to Digital Converter (ADC) at the receiver side. An infinite number of MIMO channels are used in wireless networks to improve efficiency with Cross Entropy Optimization (CEO). ADC is a serious issue because the data of the accepted signal are completely lost. ADC is used in the MIMO channels to overcome the above issues, but it is very hard to implement and design. So, an efficient way to enhance the estimation of channels in the MIMO system is proposed in this paper with the utilization of the heuristic-based optimization technique. The main task of the implemented channel prediction framework is to predict the channel coefficient of the MIMO system at the transmitter side based on the receiver side error ratio, which is obtained from feedback information using a Hybrid Serial Cascaded Network (HSCN). Then, this multi-scaled cascaded autoencoder is combined with Long Short Term Memory (LSTM) with an attention mechanism. The parameters in the developed Hybrid Serial Cascaded Multi-scale Autoencoder and Attention LSTM are optimized using the developed Hybrid Revised Position-based Wild Horse and Energy Valley Optimizer (RP-WHEVO) algorithm for minimizing the “Root Mean Square Error (RMSE), Bit Error Rate (BER) and Mean Square Error (MSE)” of the estimated channel. Various experiments were carried out to analyze the accomplishment of the developed MIMO model. It was visible from the tests that the developed model enhanced the convergence rate and prediction performance along with a reduction in the computational costs.

## 1. Introduction

MIMO is an internet-based transmission technique that uses numerous antennas for transmitting and receiving information [1,2,3,4,5,6]. The demand for wireless production with excellent Quality of Service (QoS) is increasing day by day [7,8,9]. Numerous antennas are used in both the transmission and reception sides of the wireless network to reduce errors, optimize the data speed, and improve the capacity of the transmission [10]. Wireless data traffic and the credibility of the system are handled by using the Millimeter Wave (mmWave) and massive MIMO communication system [11]. “Orthogonal Frequency Division Multiplexing (OFDM)” technologies are used in wireless networks for transmission of data. Here, a single information stream is divided into multiple sub-channel frequencies [12]. In a multi-path propagation system, OFDM provides a high standard of interaction and improves the spectrum productivity of the system [13]. When the same information is received through more than one path, interferences of the signal may occur, and these signals are distorted by many factors such as huge obstacles and multi-path propagation [14]. Machine learning is commonly used in recent wireless communication works. Channel estimation [15], resources management, signal encoding, and decoding, as well as security purposes, are some of the applications of MIMO with OFDM systems [16].

Bridge nodes are introduced in the communication path to increase the proposition and speed of the wireless network, yet the Packet Delivery Ratio (PDR) is low [17]. In wireless networks, the sensor node is one of the main components for data acquisition, buffering, and caching the data [18]. It is capable of self-testing, monitoring, and forwarding the data packets without any delay. MM-Wave is used in wireless networks for producing high-quality information [19]. The wireless image sensor network is implemented using low power and low cost, and it is commonly used in applications such as surveillance systems and environment issues monitoring systems [20]. The efficiency of the underwater channel is improved by utilizing a trained Neural Network model with the pilot signal. It produces effective outcomes when compared to linear MMSE. The interrelationship efficiency of the model is enhanced by utilizing the Channel State Information (CSI) from all the antennas present in the Base Station (BS). The channel assessment is performed to ensure that a particular channel is not used by other devices. Adaptive channel estimation is used in digital communication [21] for estimating the information using linear channel estimation techniques. Phase shifter with Radio Frequency (RF) is used in both the transmitter and receiver side of the antennas to reduce the hardware complexity and computation cost. 

In the MIMO system, multiple antennas are used for sending and receiving signals, which addresses the contamination of the channel signal and the increase in the data rate without increasing the bandwidth of the system [22]. The capacity of the system linearly increases with numerous antennas connected to the output side [23]. Channel estimation is one of the main research topics in the communication system [24]. In this proposed MIMO model, a serial cascaded multi-scaled autoencoder is used to predict the model coefficient based on the receiver-side error ratio [25,26,27,28]. The main contribution of the proposed MIMO channel estimation model is listed below. MIMO systems are considered to be very sensitive to channel matrix probability and antenna interconnection [29]. A digital signal processing chip is needed for calculating mathematical algorithms. These chips collect real-time data such as pressure, temperature, audio, or video signal and then manipulate them to improve the efficiency of the system [30,31,32,33,34,35,36]. The hardware complexity and resource necessity are complex when compared to a single antenna-based system. However, it provides high-speed communication in wireless networks without increasing the transmission power and bandwidth [37,38]. The MIMO system allows multiple user interactions simultaneously without any delay. In MIMO, a complex signal processing method is needed to assist multiple antennas at the receiver side. Assembling the antennas in the MIMO system is a time-consuming process when compared with the traditional antenna design [39]. The information stored in this network can be easily hacked and compressed by external means, which leads to the mislaying of the information [40,41,42]. Therefore, we have developed a new model for estimating the channel in MIMO using a serial cascaded deep network aided by an optimization strategy.

The contributions of the developed hybrid intelligent MIMO channel estimation scheme to predict the coefficients of channels in wireless networks are mentioned below.

A deep learning-based channel estimation model is developed to estimate the channel coefficient in the MIMO system at the transmitter side by reducing SNR in the receiver side.The Hybrid RP-WHEVO algorithm is developed for optimizing the parameters from the autoencoder and LSTM to boost the efficiency of the MIMO system during channel assessment.An HSCN is developed for the determination of channel coefficients in MIMO, where the attention LSTM and autoencoder are used. The parameters in the HSCN are optimized using the RP-WHEVO algorithm in order to minimize the RMSE, MSE, and BER, and hence the spectral capability of the system is improved.The efficiency of the channel is ensured by comparing the execution of the developed model with various optimization algorithms and traditional channel estimation techniques in regard to several error metrics.

The remaining parts of the initiated MIMO channel estimation system using hybrid deep learning are listed below. Section 2 explains the features and challenges of the existing models. The description of the MIMO system model and its implementation steps are described in Section 3. Structural representations of serial multi-scale attention networks and optimization algorithms are explained in Section 4. Section 5 describes the estimation of the channel coefficient in MIMO using deep learning technique. The resultant analysis and conclusion are described in Section 6 and Section 7.

## 2. Literature Survey

### 2.1. Related Works

In 2007, Changyong et al. [1] implemented a new pattern of MIMO-OFDM wireless channels with frequency division multiplexing. The bandwidth capability of the model is improved using a blind channel estimation technique based on sub-space approaches. It was a computer-based approach developed especially for noise reduction. Blind channel approximation was accomplished by estimating the resource information from the medium. It reduced the overhead loss and BER in the channel. Virtual carrier networks with cyclic prefixes were used in this proposed model, thereby increasing the transmission channel efficiency. The experimental provocation in this approved pattern revealed the higher lifespan provided by the implemented scheme in the wireless network than numerous traditional methods.

In 2019, Hua et al. [2] designed a hybrid MIMO system with mmWave in the wireless transmission system. In this hybrid model, a large-scale array was used in both transmitter and receiver antennas to reduce the computation cost, operational complexity, and SNR produced during the retransmission of signals. In this proposed mmWaves MIMO model, a Deep Convolution Neural Network (DCNN) was used to achieve a spatial correlation of the channel. The initiated pattern spontaneously selected the suitable weight of the channel for the training process. Numerical results showed that DCNN in the wireless network attain high performances similar to MSME. 

In 2021, Javaid et al. [3] developed a massive MIMO channel estimation model for sending the data from BS to different substations. Large-scale antennas were used in this advanced model to improve the capacity of the wireless model, but hardware complication was high. Waveform Coding Technique (WCT) was used to minimize the energy error of the given transmission bit rate. Phase and amplitude estimation models were used to measure the uncertainty reduction of the system. Baseband pre-coding matrices were estimated using the best-first search method. The Signal to Interference noise Ratio (SINR) and productivity of the hybrid model were improved by comparing the resultant outcomes.

In 2020, Yudi et al. [4] suggested a channel estimation protocol using the MIMO system in a wireless network. Analog input signals were converted into digital signals using the ADC connected to the receiver side of the network. A Generative Adversarial Network (GAN) was used in this proposed model to predict the efficiency of the channel. It was provided with an anti-interference ability to improve the transmission rate among multiple users. The GAN network provided more trained data; thus the performance of the neural network was improved and obtained high robustness in massive MIMO neural network systems during channel estimation. The experimental outcomes produced smaller intermissions and consumed only a small amount of energy. This improved model has a long lifespan and high dependability for wireless networks.

In 2020, Junta et al. [5] proposed a hybrid MIMO system for the transmission of signals. A message-passing algorithm known as Belief Propagation (BP) was used for receiving and separating the received signals. The MIMO channel consisted of multiple loops, which reduced the convergence rate of the BP in wireless networks. The performance of the model was improved using the damped BP model in the network. This damped BP used the average of continuous messages with the help of weighting factors. For training the network, deep neural-based Damped Belief Propagation was implemented. The damping factor varied based on different channel correlations. Thus, the detection performances during the training and testing period were reduced. These drawbacks were overcome by using BP with the node selection method, and it was implemented to reduce the number of loops in the wireless network. Thus, the BER was reduced due to the mismatches of channel layers.

In 2020, Jae et al. [6] suggested a pattern of MIMO with SNR feedback. The main focus of this developed pattern was to appraise the coefficients in the channel using received SNR information and to reduce the static error by taking the average of squared differences between predicted values and observed values. A Recurrent Neural Network (RNN) was built in CNN to minimize the transmission distance for improving communication in wireless networks. In this proposed model, two types of fading were used to minimize the mean square error: the time-varying fading system and a quasi-static varying system. The signal strength was reduced due to the usage of various variables. The resultant outcomes were balanced with traditional techniques to illustrate the performances and effectiveness of the developed model to prove its efficiency.

In 2021, Ha et al. [7] demonstrated the channel estimation model in wireless networks for amplifying the communication reliability and reducing the computation cost of the network. The least square method was used in channel estimation to find the suitable set of data for the network and predict the performances of dependent variables. However, these methods produced high estimation errors. The error rate of the system was reduced using hybrid deep learning such as the Fully Connected Neural Network (FCNN) model. FCNN consists of a series of layers in which various neurons are connected in a single layer. The Doppler Effect was used in MIMO with multipath channels to retrieve 5G networks. It changed the frequency of the signal during the motion between the source signal and the user. The efficiency of the hybrid LSTM was high in the implemented channel estimation model.

In 2018, Chang et al. [8] developed a new model for channel estimation using deep learning. In this MIMO system, multiple numbers of antennas were used on the transmitter side, and the length of the signal was also high. Deep learning-based signal estimation and data estimation were the two types of estimation used in this wireless network. Two-layer Neural Networks (TNN) and DNN were used in this hybrid network to maintain the reliability of the system. Antennas in this wireless network were connected to the base station for transmission and receiving the signal. The resultant outcome was compared with various channel estimation methods to verify the estimation of the network model.

#### 2.1.1. Comparison with the Contribution of Prior Works

In 2023, Chen et al. [36] developed a new channel estimation protocol to estimate the cascaded channels. It has permitted the development of the channel estimation problem as a sparse recovery issue by Compressive Sensing (CS) approaches. Additionally, the authors have implemented a two-step, multi-user, joint channel estimation process. Initially, general column-block sparsity was used, and the received signals were moved onto the general column subspace. Then, row-block sparsity of the projected signals was used to develop a multi-user joint sparse matrix recovery algorithm. In the end, the experimentation was conducted to reveal the effective output. In the given research work, the authors developed a deep structure-based channel estimation approach to validate the channel coefficient of the MIMO system for reducing the SNR. The RP-WHEVO algorithm was implemented for tuning the variables from LSTM and the autoencoder to enhance the efficacy of the MIMO system. Here, an HSCN was introduced for determining channel coefficients in MIMO. The HSCN was tuned using the RP-WHEVO algorithm for deducing the MSE, BER, and RMSE. Finally, the experimentation was conducted to show the elevated performance of the offered approach. It was revealed that the designed method attained a low error rate and also enhanced the significance.

#### 2.1.2. LSTM for Channel Estimation in MIMO from Recent Approaches

In 2023, Lipsa Dash and Anand Sreekantan Thampy [41] implemented a fused RNN-LSTM network for channel estimation. Here, the constraints of the recommended RNN-LSTM were trained and selected using a hybridized Particle Swarm Optimization (PSO)-Adam optimizer. Specifically, the present channel response was estimated by the developed PSO-Adam optimizer-based RNN-LSTM model. In the end, the evaluation of the complexity of the developed model was validated using Minimum Mean-Square Error (MMSE) and Least Square (LS). However, the major issues in the MIMO systems, such as higher BER, lower spectral efficiency, and lower noise resistance, were resolved in this work. It was required to implement a robust learning scheme to further reduce the BER, which helped to enhance the ability of the developed system. In our research work, the main aim was to reduce the BER and also improve the spectral efficiency. Here, a hybridized RP-WHEVO algorithm and HSCN were introduced to resolve the above-mentioned challenges. Additionally, the parameters of the autoencoder and LSTM model were optimized with the purpose of enlarging the efficacy of the MIMO system.

### 2.2. Problem Statement

Channel assessment is needed in the MIMO systems to improve energy as well as spectral efficiency. The MIMO system leads to different challenges, such as an increment in voltage causing the change in the digital output, complete loss of the amplitude information of the received signal, and hardware complexity resulting in high-resolution ADC on the receiver side. Therefore, deep learning techniques have been developed in MIMO channel estimation using cascaded multi-scale networks. Estimation of channel condition is essential for many reasons. The accurate estimation and prediction help to improve the performance, such as improved video streaming, reduced energy consumption, and better scheduling. Many different approaches have been introduced over the past two decades. Following that, in our research work, intelligent approaches were introduced for channel estimation. The challenges and features of various existing channel estimation models in MIMO models are given in Table 1. The Blind channel estimation technique reduces the overhead loss and BER in the channel and decreases the delivery delay. However, the energy consumption is a little high and the life duration is less when compared to other networks. DCNNs have productively decreased the packet overhead and network flexibility, and during the transmission of data in wireless networks, dead nodes are easily identified. The dependent outcomes and the computation cost are high. Large-scale antennas give high production and stable network lifetime; however, it reduces the quality of the network, and the BER is high. GAN provides the long-range communications possible in this wireless network and generates artificial data that are very similar to real data. Still, the energy consumption is very problematic, and the accomplishment cost is increased. BP’s dependability path and data communication time in the network are low. However, it does not contain a fault tolerance mechanism, and amplitude information is completely lost in the network. RNN and CNN have a reduced transmission distance to improve communication in a wireless network and Packet information dropping is reduced using an RNN network. However, long sequences of data are difficult to access, and data traffic is high in this network. FCNN avoids multiple data and reduces sensor-optimized energy usage, and the size of memory is sufficient. However, data security is one of the challenging tasks. TCNN and DNN produce high power, reduce the collision in the network, and increase the service quality; however, overfitting problems and network mobility are high. The above-mentioned challenges are reduced by implementing a new channel estimation model using serial cascaded deep learning.

## 3. MIMO System Model and the Implementation Steps of Channel Estimation

### 3.1. MIMO System Model

In the MIMO system, the signal is transmitted through more than one antenna and received on multiple antennas. It consists of a number of antennas connected in both the transmitter and receiver side that is represented as $A_{z}$ and $A_{x}$, correspondingly. Trigger in the signal can be reduced by using the subcarrier $S_{er}$ signal. The input data from the transmitter antenna is denoted as $Z^{th}$ and is transferred to the receiver side antenna in S×1 vector. The cyclic prefix function is performed with the aid of the Fast Fourier Transform (FFT) based on the length of the vector and it is denoted by $L_{1}$. The vector length varies based on the channel size, which is given by $L_{1}\geq N_{1}-1$. Here, $N_{1}$ denotes the total size of the channels in the MIMO system. The cyclic prefix function is represented by $V^{y}\left(w\right)$ and it is evaluated using the formula provided in Equation (1).
(1)$$V^{y}\left(w\right)=\sum_{z=1}^{q_{t}}E_{y_{w}}^{y,z}\Gamma^{e}F^{y}\left(w\right)+\phi \left(w\right)$$

In wireless networks, MIMO has the capacity to improve the channel throughput with multiple numbers of antennas. MIMO systems improve the bandwidth of data without any transmission power. Thus, the authenticity of the system is high. The MIMO antennas operate at the same frequency by reducing the BER to improve the capability of the channel.

The impulse function is represented by $E^{y,z}$, and it is evaluated using the formula as given in Equation (2)
(2)$$E^{y,z}=\Gamma^{e}diag\left\{\sqrt{S\Gamma }{\left[E^{y,z},0_{1w\times \left(S-M\right)}\right]}^{i}\right\}\Gamma$$
(3)$$V^{y}\left(w\right)=\sum_{z=1}^{q_{t}}diag\left\{\sqrt{S\Gamma }{\left[E^{y,z},0_{1w\times \left(S-M\right)}\right]}^{i}\right\}\times V^{y}\left(u\right)+⊕\left(w\right)$$

In wireless networks, power consumption is reduced to improve the signal range and allow the routers to communicate with multiple users. The systemic presentation of the MIMO system is given in Figure 1.

### 3.2. Implementation Procedure

In the MIMO system, channel assessment is essential to estimate the channel parameters from the receiver side antennas using pilot symbols. It is designed to simultaneously reduce the overlapping of frequency and allow multiple channel estimation at the same time. Channel estimation provides useful information for future processing of signals. The developed MIMO channel estimation model undergoes a certain implementation procedure, which is explained below.

The parameters used in the implemented MIMO channel assessment model are represented as time representation, subcarrier count, and regulation order.In the MIMO system, parallel data are connected to the antennas at the receiver side. These data are selected based on arbitrary functions Tz(G) in MATLAB.The input data and carrier signal are fed to the amplitude modulator for modulating the signal.The coefficient of the carrier signal is represented as the pilot signal. These signals are occupied based on diverse baseband algorithms and equilibrium access.If no data are sent through the transmission channel during communication, then loss of the signal occurs, which leads to a change in the time waveform. Inverse FT is applied in the time waveform to improve the efficiency of the system.The Rayleigh channel model with impulse signal $Lz\left(T\right)=\left[Lz\left(0\right),\ldots ,Lz{\left(t-1\right)}^{e}\right]$ is represented as a system model. Here, t denotes a random variable.The vacant space is removed at the acceptor side using a demodulation process to improve the efficiency of the estimated channel.

In MIMO, numerous antennas are used in the center and destination side of the antenna. The transmitter antenna sends the signals and is received by the receiver antenna. Then, these signals are combined at the receiver side in order to accomplish the error and reduction of capacity gain. The received data symbol at the receiver has transmitted symbols, the channel matrix, and noise vectors. The accurate value of the forwarded signal can be estimated through the detection methods. This can be attained via the cancellation of the unwanted signals and then determined using the desired subcarrier for the entire independent transmitter. In order to reduce the noises and distortion from the received signal, channel estimation is needed. The system is pre-trained to estimate the channel efficiency using the Minimum Mean-Square Error (MMSE) method, which needs the matrix of channel correlation and the received noise power. The channel estimation error is analyzed using these trained matrices. From these channel estimation methods, the appropriate variations in the channel are obtained.

## 4. Proposed Hybrid Serial Cascaded Network for the Estimation of Channel State Information in MIMO with Hybrid Optimization Algorithm

### 4.1. Basic Autoencoder

The autoencoder and LSTM networks are used to evaluate the system coefficients in the MIMO system. In deep learning, an autoencoder [34] is used to obtain feature vectors by utilizing an “encoder-decoder module” in which the encoded signal is reconstructed by means of a decoder, and the encoder in the network compresses the input signal. Decoding is carried out to reconstruct the compressed data to their original input signal. The autoencoder consist of an input layer, output layer, and hidden layer. The encoder and decoder are the two stages of operation performed in the autoencoder structure. Initially, the input is fed to the encoder with low dimension and then to the decoder structure. The decoder reproduces the initial data. The measurements of the data are decreased by training the output layer and then fed to the next hidden layer. Here, the input data are represented by W, and the encoded result is determined using the formula in Equation (4).
(4)$$Rf=y\left(c\right)=\beta \left(EC+n_{c}\right)$$

In Equation (4), the term β represents an activation function. The input signal is again reconstructed using the decoder as given in the formula provided in Equation (5).
(5)$$C_{1}=h\left(w\right)=\alpha \left(EU+n_{u}\right)$$

In Equation (5), the term α denotes the decoder’s sigmoid activation function, and the terms $n_{c}$ and $n_{u}$ indicate the biasing vectors of the decoder variables. The decoder reconstructs the signal back to its original signal during reconstruction, hence information loss will occur.

The autoencoder must be trained to reduce the information loss using the variables $\phi =\left(E,n_{u,}n_{c}\right)$, and this function is given in Equation (6).
(6)$$\eta =\underset{\phi }{\min}A\left(C,C^{\prime }\right)=\underset{\phi }{\min}A\left(C,h\left(g\left(c\right)\right)\right)$$

Random and continuous information loss can be measured using the cross-entropy function and by square error value, respectively. These functions are given by Equation (7).
(7)$$A\left(\phi \right)=\sum_{o=1}^{m}\left[c_{i}\log\left(u_{i}\right)+\left(1-c_{i}\right)\log\left(1-u_{i}\right)\right]$$

The autoencoder reduces the dimensionality of the signal. Anomaly detection of images is possible using an autoencoder in the network. BER and signal noises are reduced by using an autoencoder system. 

### 4.2. Basic Attention LSTM

Attention-based LSTM [35] consists of two main parts: the attention model and LSTM. The attention network in LSTM selects only the useful information at a time sequence from the input signal, and their weights are assigned to reduce the overload in the signals present in its hidden state. LSTM in the network uses sequential data as input and stores information in the memory for future use. Thus, the LSTM provides an association between the preceding and succeeding data. In this wireless communication network, LSTM has the potential to forecast future data using previous information. Managing and tracking the information in LSTM is carried out using three gates, namely, a “forget gate, input gate, and output gate”.

The input data consist of related and unrelated information. Unrelated information in the network creates signal loss during communication. The signal loss is reduced by using the forget gate in LSTM, which helps to eradicate this unrelated information and the processing. The forget gate is defined by means of Equation (8).
(8)$$G\left(y\right)=\beta \left(E_{g,y}{\bar{c}}_{y}+E_{g,y}{\bar{j}}_{y-1}+n_{g,y}\right)$$

In Equation (8), the term β represents the sigmoid function, and the biasing term of the forget gate is denoted by $n_{g,y}$. If the forget gate produces output 1, then the unrelated information is dropped from the input data. Only the useful information from the input gate is fed to the next update gate.

The update gate restores the present state of the data using the previous state. The computation of the update gate is given by Equation (9).
(9)$${\bar{V}}_{t}=\tan\left(E_{v,{\bar{c}}_{y}}+E_{v,{\bar{c}}_{y-1}}+{\bar{n}}_{g,y}\right)$$
(10)$$V_{y}=g_{y}\cdot V_{y-1}+o_{y}\cdot {\bar{V}}_{y}$$
(11)$$Out=\sigma \left(E_{p}h\left(j_{y-1},c_{y}\right)+n_{p}\right)$$

In Equations (9)–(11), the terms $V_{y}$ and $V_{y-1}$ denote the current and past status of the memory and $n_{p}$ indicates the biased term. Finally, the output is generated using the tan function and threshold value to achieve the essential operation. However, autoencoders are highly sensitive to input errors. Hence, an autoencoder with a serial cascaded structure is used in this hybrid network to reduce the repetition of the input signal and minimize the number of errors in the received information. The attention mechanism used in the autoencoder to estimate the components and assign them based on their weights. A linear transformation is performed in the attention module to initiate the input signal, and this input signal is resized using a fully connected layer in the network. The input signal is split into three sub-signals: query, key, and value. These sub-signals are represented in Equation (12).
(12)$$L,U,B\in T\left(\frac{mg}{2}\right)\times 2\times M_{head}$$

In Equation (12), the term $M_{head}=B_{pilot}$ represents the size of the head. A scaled dot-product attention mechanism is performed, and the integrated result is conveyed to the next fully connected layer. Finally, the attention module output is generated using the softmax function, and it is determined using Equation (13).
(13)$$Atten=Sof\left(\frac{WL^{y}}{\sqrt{\frac{K_{g}}{2}}}\right)B$$

The adaptability between the query and the key vector is computed using a feed-forward network, and their comparability is measured using the dot-scale attention mechanism. Long input sequence data are handled by using an attention network and the interpretability of the model is improved by selecting the relevant part from the input using the attention mechanism.

### 4.3. Developed HSCN for Estimating Channel

A new channel estimation model is implemented for the MIMO system by using a cascaded deep network with a hybrid optimization algorithm. Various traditional methods are used for estimating the channel coefficient, but they produce large computation complexity and are highly time-consuming. So, a serial cascaded multi-scale autoencoder with attention to LSTM is used to eliminate the BER and reduce the time needed for the computation process of channel coefficients. The main drawback of using an autoencoder system is the overfitting problem and the need for additional training. Due to this, the MIMO system becomes overloaded, and errors are prone to occur on the receiver side. The system complexity and the overfitting problem are reduced using the hybrid serial cascaded multi-scale network. Here, the autoencoder is constructed with three components—encoder, code, and decoder—and then trained to select the input automatically. The selected input is applied to the encoder. The encoder compresses the input data using code and then again reconstructs the input data using a decoder and stores them in the memory of the receiver antenna. LSTM is used to focus on a particular feature at a time by avoiding the remaining features. The LSTM is used because of its ability to minimize the vanishing gradient problem. Here, the HSCN network is constructed to identify the channel in MIMO, where the autoencoder and LSTM are serially integrated to build a HSCN. The hidden neuron count and epochs from the autoencoder and LSTM network are optimally tuned to increase the channel assessment performance. The efficiency of the system is improved by optimizing the variables using the proposed RP-WHEVO algorithm by considering maximum iteration as 50, population counts as 10, and length of chromosome as 4. Thus, the RP-WHEVO-HSCN system effectively reduces the MSE, RMSE, and BER and generates high performance in channel estimation tasks. Channel estimation using the HSCN system is diagrammatically illustrated in Figure 2.

## 5. Estimated Channel Coefficients in MIMO System Using Deep Learning for Minimizing Bit Error Rate

### 5.1. Estimated Channel Coefficients

MIMO system in wireless communication undergoes least-square-based calculation for obtaining proper channels in the network. The original data in the network is represented by $F^{y}\left(w\right)=O^{y}\left(w\right)+J^{y}\left(w\right)$ which is made of imaginary and real parts for the inconsistent operation. The term $V^{y}\left(w\right)$ is determined using the formula given in Equation (14).
(14)$$V^{y}\left(w\right)=\sum_{z=1}^{q_{t}}diag\left\{F^{y}\left(w\right)\right\}HE^{a,z}+\Theta^{y}\left(w\right)$$

In Equation (14), the value of the $V^{y}\left(w\right)$ matrix becomes reduced by substituting the $F^{y}\left(w\right)$ function. The reduced matrix is computed using Equation (15).
(15)$$V^{y}\left(w\right)=\sum_{z=1}^{q_{t}}\left(diag\left\{O^{y}\left(w\right)\right\}+diag\left\{J^{y}\left(w\right)\right\}\right)\times HE^{a,z}\Theta^{y}\left(w\right)$$

The complexity of the system can be reduced by changing the parameter $\sqrt{S\Gamma }$ in FFT. The output obtained by means of carrying out FFT is given by Equation (16).
(16)$$V_{y}\left(w\right)=\sum_{z=1}^{q_{t}}O_{diag}^{y}\left(w\right)HE^{a,z}+J_{diag}^{y}\left(y\right)HE^{a,z}+\Theta^{y}\left(w\right)$$

The MIMO system is trained to reduce the BER using a time index o∈{0,1,…,k−1}, and the trained data are represented with the aid of Equation (17).
(17)$$V^{w}=TE^{w}+YE^{w}+\Theta^{y}\left(w\right)$$

The variable $V^{w}$ is generated by utilizing the G and F matrices and the term $E^{a,z}$ indicates a circular matrix. The interpretations for the matrices are given in the below Equations (18) and (19).
(18)$$G=\left[\begin{array}{ccc} P_{diag}^{1}\left(0\right)E & \cdots & P_{diag}^{d^{a}}\left(0\right)E\\ \vdots & \cdots & \vdots \\ P_{diag}^{1}\left(k-1\right)E & \cdots & P_{diag}^{d^{a}}\left(k-1\right)E \end{array}\right]$$
(19)$$F=\left[\begin{array}{ccc} J_{diag}^{y}\left(0\right)E & \cdots & J_{diag}^{d^{o}}\left(0\right)E\\ \vdots & \cdots & \vdots \\ J_{diag}^{y}\left(k-1\right)E & \cdots & J_{diag}^{d^{o}}\left(0\right)E \end{array}\right]$$

The inverse $Y^{+}$ and least square value of the matrix is calculated using the formula provided in Equation (20).
(20)$$Y^{+}={\left(Y^{E}Y\right)}^{-1}Y^{E}$$
(21)$$\hat{E}y^{w}=E^{w}+Y^{+}TV^{w}+Y^{+}\Theta^{y}\left(w\right)$$

The value $\hat{E}y^{w}$ in Equation (21) is constructed by grading the vector values that lead to system noise. The clarity is determined using Equation (22).
(22)$$\hat{E}y^{w}=Y^{+}V^{w}=E^{w}+Y^{+}\Theta^{y}\left(w\right)$$

The value of the matrix Y is derived by using the variables $\Theta^{y}\left(w\right)$ and $V^{w}$, from the above equations. The matrix Y is determined using Equation (23).
(23)$$Y=\left[\begin{array}{c} \begin{array}{ccc} J_{diag}^{1}\left(0\right)\tilde{H}\left(0\right) & \cdots & J_{diag}^{d^{y}}\left(0\right)\tilde{H}\left(0\right) \end{array}\\ \begin{array}{ccc} \vdots & \cdots & \vdots \end{array}\\ \begin{array}{ccc} J_{diag}^{y}\left(k_{1}-1\right)\tilde{H}\left(k-1\right) & \cdots & J_{diag}^{d^{y}}\left(k_{1}-1\right)\tilde{H} \end{array}\left(k_{1}-1\right) \end{array}\right]$$

The diagonal matrix has a non-numeric value and it is denoted by $J_{diag}^{1}$.

### 5.2. Objective Function of Developed Channel Estimation

The implemented channel estimation model using deep learning greatly reduces the BER, RMSE, and MSE. The HSCN network with a hybrid RP-WHEVO algorithm is proposed for estimating the variables to predict the channel coefficient in the MIMO system at the transmitter side by reducing SNR. Variables such as “hidden neuron count and count of the epoch” of the autoencoder and LSTM network are optimized using the RP-WHEVO algorithm in order to improve the spectral efficiency by reducing RMSE, MSE, NMSE, and BER. The main intention of the proposed channel estimation scheme aided by a heuristic-based deep learning approach is given in Equation (24).
(24)$$K=\underset{\left\{VL_{l}^{AE},BS_{r}^{AE},VL_{t}^{ALSTMN}BS_{d}^{ALSTM}\right\}}{\arg\min}\left(\begin{array}{l} 0.5*\left(RMSE+MSE\right)\\ +0.5*BER+NMSE+\frac{1}{SE} \end{array}\right)$$

In Equation (24), the terms $VL_{l}^{AE}$ and $VL_{t}^{ALSTMN}$ denote the tuned neuron counts that are optimized in the range [5−225] in both the autoencoder and LSTM network, and the terms $BS_{r}^{AE}$, $BS_{d}^{ALSTM}$ denote the tuned count of epochs that are optimized in the range [5−50] in both the autoencoder and LSTM network and the term SE denotes the spectral efficiency. The formula for computing the values of RMSE, MSE, NMSE, and BER are given by Equation (25), Equation (26), Equation (27) and Equation (28), respectively.
(25)$$RMSE=\sqrt{\sum_{l=1}^{m}\frac{{\left(XZ-VC\right)}^{2}}{G}}$$
(26)$$MSE=\frac{1}{G}\sum_{l=1}^{G}{\left(XZ-VC\right)}^{2}$$
(27)$$NMSE=\frac{\sum_{l=1}^{m}{\left(XZ-VC\right)}^{2}}{\sum_{l=1}^{m}{\left(VC\right)}^{2}}$$
(28)BER=RebitTobit

Spectral efficiency is defined as the total amount of information transmitted per unit bandwidth. Here, the terms XZ and VC represent the predicted value and actual value, respectively. The term Rebit denotes the number of bits at the receiver side and $To_{bit}$ denotes the total number of bits.

### 5.3. Proposed RP-WHEVO

The initiated RP-WHEVO algorithm is used in this hybrid serial cascaded attention network for optimizing the parameters in the autoencoder and the LSTM to improve the spectral coherences of the MIMO system by decreasing the RMSE, MSE, NMSE, and BER of the system. Variables such as hidden neurons and epoch count are optimized using the developed RP-WHEVO algorithm from both the autoencoder and LSTM network. Thus, the proposed RP-WHEVO algorithm optimizes the variables to increase the spectral productivity of the MIMO system during channel estimation. The WHO algorithm can easily fall into local optimal problems that slow down the searching speed of the algorithm, and parallel computation is also a difficult task in the conventional WHO algorithm. These issues are reduced by developing a hybrid and new optimization technique called the RP-WHEVO algorithm. In this RP-WHEVO algorithm, the variable y is newly introduced for updating the solutions using WHO and EVO algorithms. If the condition $\left(y<2*\frac{\max iter}{4}\&\&y\geq \frac{\max iter}{4}\right)$ or $\left(y<3*\frac{\max iter}{4}\&\&y\geq 2*\frac{\max iter}{4}\right)$ is satisfied, then the solution is updated using the WHO algorithm, or else the position is updated using EVO. Here, the term y denotes the randomly selected parameter and maxiter denotes the maximum number of iterations. Finally, the best position is obtained from the procedure of the hybrid RP-WHEVO algorithm.

Wild Horse Optimization: This algorithm is proposed to overcome certain challenges, such as high cost in computation and heavy load during cloud computation tasks, and has the power to reduce the local optimal problem. It consists of five main phases for optimization. The first phase starts with creating an initial population and forming a group. The second phase explains the grazing behavior of the horse. The third phase describes how a leader can lead the entire group. Finally, select the best leader and save the best solution in the problem space.

Initially, the random population is created and represented as $\bar{C}=\left\{c_{1},c_{2},\ldots ,c_{n}\right\}$ and this initial population is divided into various numbers of groups with a leader, and it is represented by Gq. The total members in the group are represented by Gq=[C×QD], where QD represents the total percentage of leaders in the group, and it is considered as the controlling parameter for the proposed algorithm.

The grazing manner of this algorithm is executed by considering the leader as a center point, and the remaining members search around the center point; the grazing is executed using the formula given in Equation (29).
(29)$$B{\bar{V}}_{o,h}^{k}=2X\cos\left(2\pi TX\right)\times \left(L-BV_{o,h}^{k}\right)+L$$

In Equation (29), the term $BV_{o,h}^{k}$ represents the member’s current position, L denotes the leader of the group, and $B{\bar{V}}_{o,h}^{k}$ the term represents an adaptive function used in the grazing mechanism. The updated position after grazing is represented by $B{\bar{V}}_{o,h}^{k}$.

The adaptive mechanism TX is executed using the formula given in Equation (30).
(30)$$TX=Y_{2}⊕AFC+Y_{3}⊕\left(\approx AFC\right)$$

In Equation (30), the terms $Y_{2}$ and $Y_{3}$ indicate the random numbers in the range of [0−1] and AFC denotes the control parameter. Initially, the values of these parameters start from 1 and gradually reduce to 0. This parameter is determined using the formula in Equation (31).
(31)$$AFC=1-py\times \left(\frac{1}{M_{it}}\right)$$

In Equation (31), the present iteration is represented by py and $M_{it}$ indicates the supreme count of iteration for executing the grazing behavior of the algorithm.

A new hybrid updated position is formed by using crossover operation, and it is computed using Equation (32).
(32)$$C_{hj}^{q}=CO\left(C_{hj}^{w},C_{hj}^{r}\right)$$

In Equation (33), the terms $C_{hj}^{w}$ and $C_{hj}^{r}$ represent the positions of host parameters, and the term $C_{hj}^{q}$ denotes the hybrid updated position. 

Leaders of the groups lead the entire group to the searching point. First, the dominant group continues their search, and the remaining group waits in a queue until the dominant group finishes their search. The searching operation is determined using the formula given in Equation (33).
(33)$$\bar{L}=\{\begin{array}{cc} 2X\cos\left(2\pi TX\right)\times \left(EG-L_{h}\right)+EJ & ifY_{3}>0.5\\ 2X\cos\left(2\pi TX\right)\times \left(EG-L_{h}\right)-EJ & ifY_{3}\leq 0.5 \end{array}$$

In Equation (33), the next position of the leader is represented by $\bar{L}$ and EG denotes the searching point position.

The performance of the algorithm is maintained by selecting the best leader for the group. The leaders are determined based on their fitness value. If any member in the group has a leading fitness value, then the locality of the leader with their respective position will interchange based on Equation (34).
(34)$$L_{\mathrm{int}}=\left\{\begin{array}{cc} C_{ho} & if\cos y\left(C_{ho}\right)<\cos y\left(L_{\mathrm{int}}\right)\\ L_{\mathrm{int}} & if\cos y\left(C_{ho}\right)>\cos y\left(L_{\mathrm{int}}\right) \end{array}\right\}$$

Thus, the best output solution is obtained by randomly selecting the position. Yet local optimal problems can easily occur, which leads to a decrease in the convergence rate. 

Energy Valley Optimizer: This is based on nuclear reaction. Here, two sub-atomic particles strike each other to produce a new particle. Most of the particles in the universe are unstable in nature. These unstable particles emit energy in a loss process known as decay. The rate of decay differs based on the particles. The low-energy particle undergoes a decaying process. In this decaying process, external energy is ejected using the emission process. One of the most challenging processes is determining the stability of the particles by utilizing the number of neutrons $Na_{1}$ and protons $Pa_{1}$ which are present in the ratio of $\frac{Na_{1}}{Pa_{1}}$. A small value is considered as the ratio for lightweight particles, and for heavier particles, the value of the ratio is assumed to be a high value. The position of neutron-rich particles moves towards the stability bond, and neutron-less particles undergo the capturing of the electron process to stabilize the neutron and then move to the stability bond.

The emission of energy undergoes three decaying processes with various levels of stability. The positively charged particles are denoted by α particles, the negatively charged particles with high-speed electrons are denoted by β particles, and the high-energy protons are denoted by γ particles.

The enrichment bonds between the neutron-rich and neutron-poor particles are determined using the formula given in Equation (35).
(35)$$NM=\frac{\sum_{o=1}^{m}MRK_{y}}{G},0=1,2,\ldots G$$

In Equation (35), the term G represents the enrichment bond and MRK denotes the neuron enrichment level of the $o^{th}$ particles.

The strength of the particles is determined based on the best and worst levels of stability, and its formula is given by Equation (36).
(36)$$StaL_{1}=\frac{MRK-VA}{ED-VA},0=1,2,\ldots m$$

In Equation (36), the terms ED and VA denote the best and worst positions of the particles. If $\left[MRB_{y}>NM\right]$ so, then the ratio of stability is high. The newly generated particle in the universe is updated based on the particle position vector, and the formula for a position update is given by Equation (37).
(37)$$Cv_{o}^{N}=Cv\left(C_{jh}\left(c_{o}^{k}\right)\right),\{\begin{array}{c} o=1,2,\ldots m\\ k=alpha \end{array}$$

The stability of the existing particle is increased by using the gamma break-up process. By updating the position in the EVO algorithm, the new solution of the candidate is updated. The distance between the excited particles and the considered particle is determined using the formula given in Equation (38).
(38)$$Dis=\sqrt{{\left(c_{2}-c_{1}\right)}^{2}+{\left(u_{2}-u_{1}\right)}^{2}},\{\begin{array}{c} o=1,2,\ldots m\\ l=1,2,3,\ldots ,m-1 \end{array}$$

In Equation (38), the terms $o^{th}$ and $l^{th}$ are the two nearest particles with the coordinates $\left[c_{1,},c_{2}\right]$ and $\left[u_{1,},u_{2}\right]$ where the total distances between them are represented by Dis. The second best position $C_{o}^{New}$ is updated using the position vector $C_{mh}$ of the nearest particle, and the formula is given by Equation (39).
(39)$$C_{o}^{New}=C_{i}\left(C_{mh}\left(C_{o}^{k}\right)\right),\{\begin{array}{c} o=1,2,\ldots m\\ k=gamma \end{array}$$

If the particles have less stability, then the beta decay process is observed to boost the durability level of the member by updating the position. The position updating process is controlled by using the best level of stability $C_{best}$ and the center of the particle $C_{cent}$ in the search space. The new movement of a particle is determined using the formula provided in Equation (40).
(40)$$C_{o}^{New}=C_{o}+\frac{\left(t_{1}\times C_{ND}-t_{2}\times C_{VQ}\right)}{DA_{o}},o=1,2,3\ldots m$$

In Equation (40), the ongoing position of the $o^{th}$ particle is represented by $C_{o}$ and $C_{VQ}$ denotes the center position. Here, the terms $t_{1}$ and $t_{2}$ indicate the two random numbers in the range [0,1]. The “exploration and exploitation phase” of the algorithm is improved by updating the best stability level and the nearest position of the particle. The EVO algorithm is executed in two loops that have three different updated positions while two positions are updated during the exploration phase, which leads to the local optimum problem, and the remaining one position is updated during the exploitation phase. The best global position of the candidate is updated. Algorithm 1 gives the pseudocode for the proposed RP-WHEVO algorithm, and the flowchart of the implemented RP-WHEVO algorithm is given in Figure 3.
**Algorithm 1:** Proposed RP-WHEVO1: Set the size of the population as G and the maximum number of iterations as Y
2: The initial population of both the EVO algorithm and WHO algorithm are initiated3: The fitness value is calculated for every solution4: Create a number of groups and assign their leader based on fitness value5: For r→1toY
6: For Q→1toG
7:
  If $\left(y<2*\frac{\max iter}{4}\&\&y\geq \frac{\max iter}{4}\right)$
8:

  Evaluate the position by means of the WHO algorithm9:
  Else10:

  Evaluate the position by means of the EVO algorithm11:
  End if12: End13: End14: Obtain the best solution15: End

## 6. Results and Discussion

### 6.1. Experimental Setup

The proposed RP-WHEVO-HSCN has been used for estimating the channel coefficient in the MIMO system used for the wireless communication process. The implemented RP-WHEVO-HSCN-based channel estimation model in MIMO was designed in the MATLAB2020a paradigm. The performance was evaluated by comparing the resultant output from the executed channel estimation model with various traditional algorithms and newly implemented channel estimation techniques. The newly developed channel estimation schema in MIMO was executed with a maximum iteration of 50, a chromosome length of 4, and a population size of 10. The performance of the implemented RP-WHEVO-HSCN-based channel estimation model in MIMO was compared with different algorithms such as Harris Hawks Optimization (HHO) [28], Reptile Search Algorithm (RSA) [29], Wild Horse Optimization (WHO) [26], and Energy Value Optimization (EVO) [27] algorithms and with recent channel estimation techniques such as Deep Neural Network (DNN) [30], Sparse Code Multiple Access (SCMA) [31], Long Short Term Memory (LSTM) [32], and Autoencoder + LSTM [33]. The simulation parameters of the designed channel estimation in the MIMO system are given in Table 2.

### 6.2. Cost Function Computation

The cost function evaluation is carried out in this developed channel estimation model in the MIMO system for estimating the coefficient of the channel in the wireless network by varying the number of iterations as given in Figure 4. From the analysis, it is seen that the cost function of the implemented RP-WHEVO-HSCN-based channel estimation model is 99.3%, 99%, 99.6% and 99.3% enhanced versus the HHO-HSCN, RSA-HSCN, WHO-HSCN, and EVO-HSCN models, correspondingly, at the iteration count of 20. From the cost function output, it is verified that the proposed RP-WHEVO-HSCN-based channel estimation model reduces the cost consequence to enlarge the effectiveness of the hybrid MIMO system.

### 6.3. Performance Evaluation of the Suggested Channel Estimation Model in the MIMO System by Considering Various Techniques

The error performance evaluation of the proposed channel estimation scheme for the MIMO system in wireless communication by varying SNR among different techniques is specified in Figure 5. The designed RP-WHEVO-HSCN-based channel estimation has a MAE that is 97.2% enhanced versus DNN, 97.3% enhanced versus SCMA, 97.32% enhanced versus LSTM, and 97.05% enhanced versus Autoencoder + LSTM at SNR 20. Thus, the MAE of the initiated RP-WHEVO-HSCN-based channel estimation model is reduced to improve the spectral efficiency of the serial cascaded-based channel assessment model. 

### 6.4. Performance Evaluation of Suggested Channel Estimation Model in MIMO System by Considering Different Optimizing Algorithms

The error performance evaluation of a wireless proposed channel estimation scheme for the MIMO system using a serial cascaded Multiscale network is specified in Figure 6. The proposed RP-WHEVO-HSCN model-based channel estimation reduces the MPE, which is enhanced by 3.2% versus HHO-HSCN, 9.0% versus RSA-HSCN, and 14.23% versus EVO-HSCN at SNR 20. Thus, the proposed RP-WHEVO-HSCN-based channel coefficient estimation model reduces the MPE value to increase the coherences of the MIMO system to predict the coefficient at the transmitter side using SNR feedback signal information at the receiver side antenna. 

### 6.5. Performance Analysis of the Implemented Channel Estimation Scheme for the MIMO System Model over Various Measures

The performance evaluation of the executed channel estimation model for the MIMO system model among various measures is given in Figure 7. The delineated RP-WHEVO-HSCN-based channel estimation in the MIMO system improved its spectral efficiency by 18.6% versus HHO-HSCN, 21.3% versus RSA-HSCN, 24.9% versus WHO-HSCN, and 24.2% versus EVO-HSCN at the SNR 15. Thus, the error rate is reduced in the proposed model to improve the spectral efficiency of the hybrid MIMO system model.

### 6.6. Performance Analysis of Over-Optimized Algorithm and Various Techniques

The performance analysis of the RP-WHEVO-HSCN-based channel estimation for estimating the channel coefficient in the MIMO system model against various optimization algorithms and existing techniques is demonstrated in Table 3 and Table 4. The RMSE of the proposed RP-WHEVO-HSCN-based channel estimation model for the MIMO system is improved by 4.8% versus HHO-HSCN, 1.4% versus RSO-HSCN, 2.6% versus WHO-HSCN, and 3.7% versus EVO-HSCN. Thus, the RMSE value of the proposed RP-WHEVO-HSCN-based channel estimation model for the MIMO system is reduced to improve the efficiency of the channel estimation scheme.

### 6.7. Statistical Report on the Implemented Channel Estimation Model

The statistical report on the newly initiated channel estimation model for the MIMO system with the aid of the hybrid serial cascaded network in a wireless network is given in Table 5. The mean value of the initiated RP-WHEVO-based channel estimation model is 2.53% more enhanced than EBOA-HSCN, 7.68% more enhanced than GMO-HSCN, 8.42% more enhanced than COA-HSCN, and 2.31% more enhanced than AVOA-HSCN for the SNR rate 5. From this statistical report, it is proved that the ability of the RP-WHEVO-based channel estimation model in the MIMO system using SNR information at the output side of the antenna is upgraded, and the BER is reduced effectively. 

### 6.8. Computation Complexity Analysis on the Offered Approach

The computation complexity of the recommended channel estimation techniques in MIMO systems is given in Table 6. Here, the variable MaxIter indicates the maximum iteration and $N_{pop}$ denotes the number of populations. 

### 6.9. Evaluation of the Performance of the Designed Channel Estimation Scheme in the MIMO System Using Recent Approaches

Evaluation of the designed channel estimation scheme in MIMO systems against recent approaches is given in Table 7. Thus, the simulation outcome of the proposed model attains better results than the other baseline approaches.

### 6.10. Validation of the Designed Channel Estimation Model Using Diverse SNR Rate

Computation of the performance offered by the proposed channel estimation in MIMO systems regarding diverse SNR rates is shown in Table 8. The experimentation on the designed RP-WHEVO-HSCN-based model shows that the proposed scheme attains an enriched performance compared to several other baseline approaches.

## 7. Conclusions

A neural network-based channel efficiency estimation scheme was developed to estimate the channel coefficient in the wireless network. The proposed system was developed to predict the channel coefficient of the MIMO system at the transmitter side antenna by utilizing the SNR feedback information to reduce the error rate. The serial cascade-forward network provides a nonlinear relationship between input and output antennas without changing the linear relationship between the antennas. The coefficient of the system was predicted by utilizing an autoencoder and LSTM network. The parameters in this network were optimized using the RP-WHEVO algorithm to achieve an improved channel coefficient in the system by decreasing the model intricacy. The results from this system were evaluated by comparing the performances of the implemented channel estimation scheme against various heuristic algorithms and different channel estimation techniques. From the experimental outcomes, it was seen that the effective achievable rate of estimating the channel coefficient by the executed channel estimation scheme in the MIMO system was improved by 15% versus HHO-HSCN, 66.3% versus RSA-HSCN, 42.8% versus the WHO-HSCN algorithm, and 23.4% versus EVO-HSCN at an SNR value of 25. Thus, the initiated RP-WHEVO-HSCN-based channel estimation offered a higher spectral effectiveness rate in estimating the channel than several other conventional techniques. In the future, the non-coherent systems [42] based on the DPSK scheme will be surely included in our proposed algorithm to show effective performance.

## Figures and Tables

**Figure 1 sensors-23-09154-f001:**
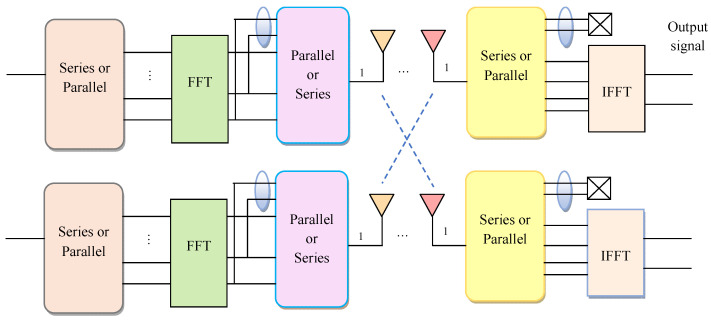
Structural representation of the MIMO system model.

**Figure 2 sensors-23-09154-f002:**
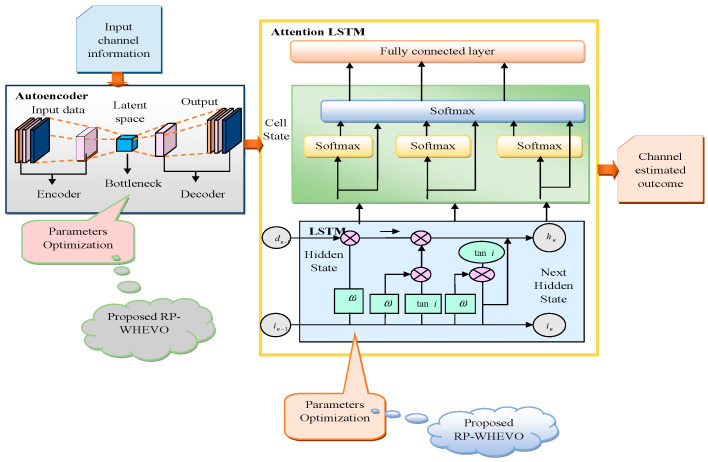
Diagrammatical illustration of channel estimation using HSCN.

**Figure 3 sensors-23-09154-f003:**
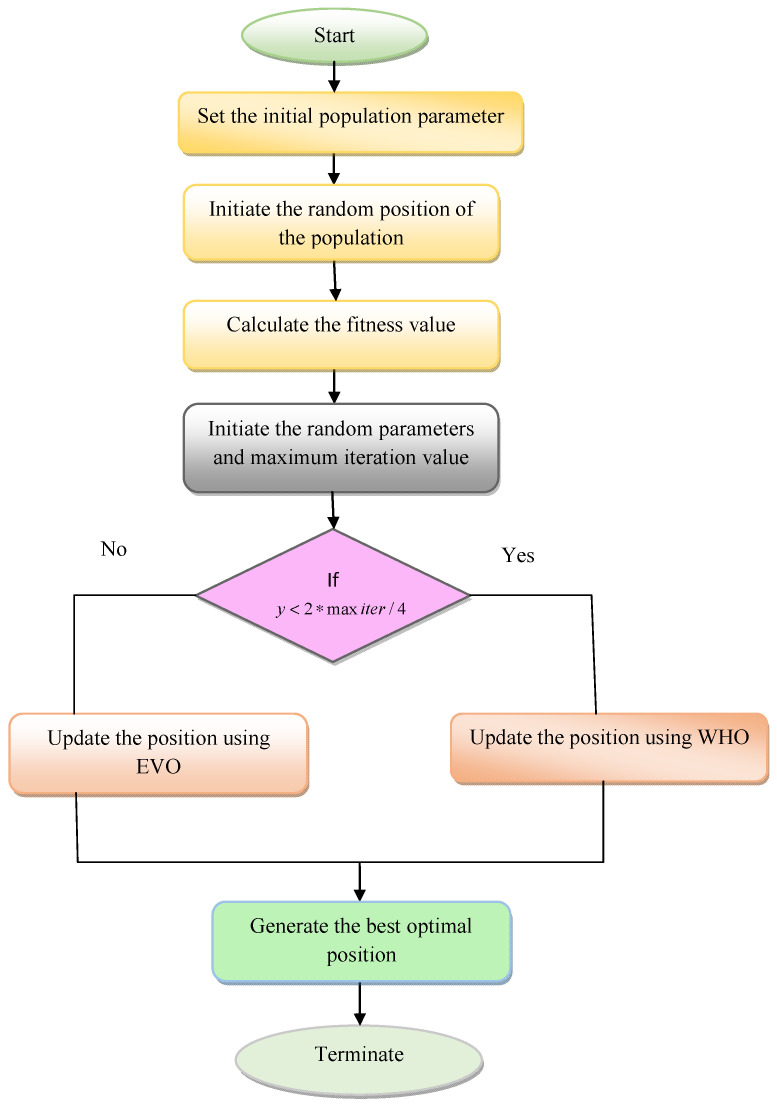
Flowchart of the proposed RP-WHEVO algorithm.

**Figure 4 sensors-23-09154-f004:**
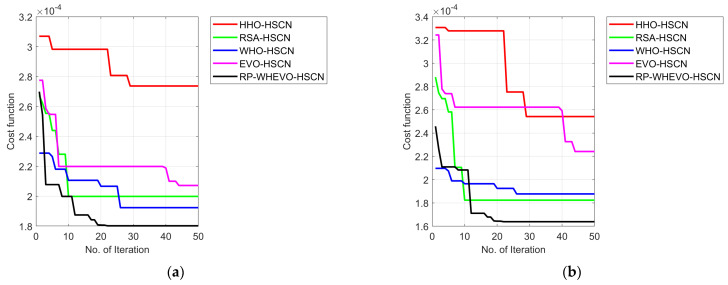
(**a**–**e**) Cost function analysis on the proposed serial cascaded deep learning-based channel estimation model in the MIMO system.

**Figure 5 sensors-23-09154-f005:**
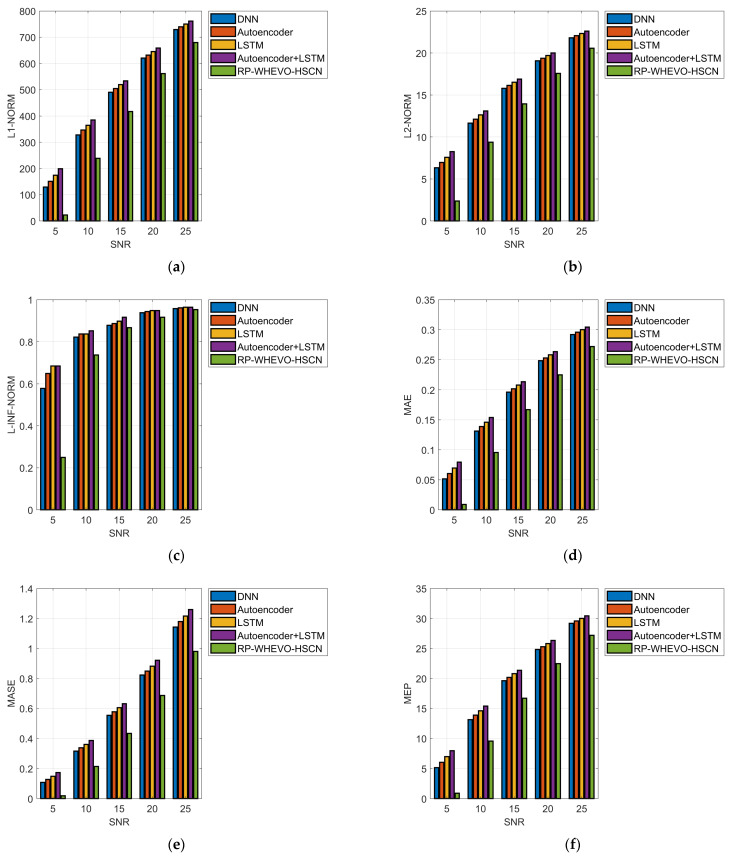
Performance assessment of the implemented serial cascaded deep learning-based channel estimation model in terms of (**a**) L1-Norms, (**b**) L2-Norms, (**c**) L-INF-NORM (**d**) MAE, (**e**) MASE, (**f**) MEP, (**g**) RMSE, and (**h**) SMAPE.

**Figure 6 sensors-23-09154-f006:**
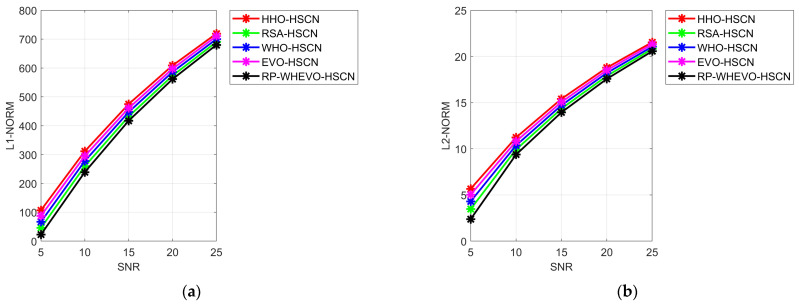
Performance assessment of the proposed serial cascaded deep learning-based channel estimation model in terms of (**a**) L1-Norms, (**b**) L2-Norms, (**c**) L-INF-NORM (**d**) MAE, (**e**) MASE, (**f**) MEP, (**g**) RMSE, and (**h**) SMAPE.

**Figure 7 sensors-23-09154-f007:**
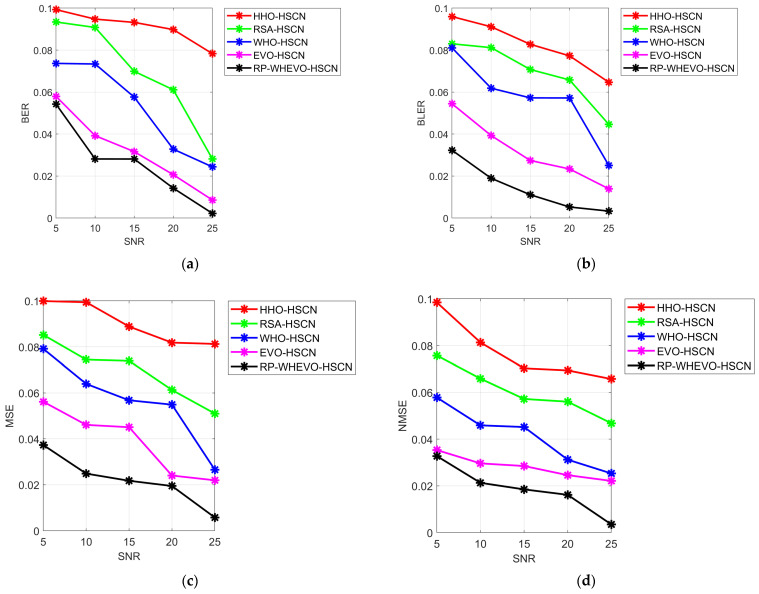
Performance assessment of the serial cascaded deep learning-based channel estimation model in terms of various measures such as (**a**) BER, (**b**) BLER, (**c**) MSE, (**d**) NMSE, (**e**) Spectral efficiency, (**f**) Sum rate, and (**g**) Effective achievable rate.

**Table 1 sensors-23-09154-t001:** Features and Challenges of the Prior Channel Estimation Techniques in MIMO Systems.

Author [Citation]	Methodology	Features	Challenges
Changyong et al. [1]	Blind channel estimation technique	Reduces the overhead loss and BER in the channel. Decreases the carriage lag.	Energy utilization is a little high. Life span is less when compared to other networks.
Hua et al. [2]	DCNN	Productively decreases packet loss and enhances network flexibility. During the transmission of data in a wireless network, dead nodes are easily identified.	Does not provide independent outcomes. Computation cost is high.
M. Chinnusami et al. [3]	WCT	Gives high production. Provides a stable network lifetime.	Reduces the quality of the network. BER is high.
Yudi et al. [4]	GAN	Long-range communications are possible in this wireless network. Generates artificial data that are very similar to real data.	Energy consumption is very high. Implementation cost is high.
Dong, P et al. [5]	BP	Path dependability in the network is low. Total data communication time is low.	Does not contain a fault tolerance mechanism. Amplitude information is completely lost in the network.
Ravindran et al. [6]	RNN and CNN	The transmission distance is reduced to improve communication in the wireless network. Packet information dropping is reduced using the RNN network.	Long sequences of data are difficult to process. Data loss and data traffic are high in this network.
Navabharat Reddy et al. [7]	FCNN	Avoids the data load and reduces the energy value of sensors. The memory size is acceptable, so it prevents the packet of information loss in the network.	Data security is one of the challenging tasks. Small-area implementation is not possible in this model.
Tachibana, et al. [8]	TNN and DNN	Power consumption is less and reduces the collision in the network. The quality of the network is high.	Network probability is high. Suffers from overfitting problems.

**Table 2 sensors-23-09154-t002:** Simulation Parameters of the Offered Channel Estimation in MIMO Systems.

Parameters	Values
Subcarrier count	128
Number of blocks in each channel realization	1
Modulation order	M = 4
OFDM sample time	1 × 10^−7^
Guard interval time	16
OFDM symbol time	128 × 1 × 10^−7^
Location of the pilot in subcarrier	[20, 30, 40, 50, 60, 70 …… 120]
Number of subcarriers that carry data	12
Channel trap count	3
Doppler in Hz	0.1
Number of columns in the dictionary	128
Channel SNR	15
Channel SNR for sweep	[5, 7, 9, 11, 13, 15, 17, 19, 21, 23, 25, 27, 29]
Iteration count	1 × 10^2^
Inner loop length	25

**Table 3 sensors-23-09154-t003:** Performance Analysis of the Implemented Serial Cascaded Deep Learning-Based Channel Estimation Model against Various Algorithms.

Performance Measures	EVO-HSCN	WHO-HSCN	RSO-HSCN	HHO-HSCN	RP-WHEVO-HSCN
MEP	28.408	28.003	27.605	28.762	27.20
SMAPE	42.311	41.518	40.765	43.018	39.994
MASE	107.53	10.435	10.127	11.011	97.987
MAE	28.408	28.003	27.605	28.762	27.22
RMSE	42.614	42.102	41.622	43.07	41.121
L1-NORM	71020	70.006	69.013	71.906	67.999
L2-NORM	21.307	21.051	20.811	21.535	20.561
L-INF-NORM	95.776	95.776	95.776	95.776	95.307

**Table 4 sensors-23-09154-t004:** Performance Analysis on the Developed Serial Cascaded Deep Learning-Based Channel Estimation Model against Various Techniques.

Performance Measures	SCMA + LSTM	LSTM	SCMA	DNN	RP-WHEVO-HSCN
MEP	30.445	30.005	29.583	29.176	27.2
SMAPE	46.406	45.505	44.658	43.842	39.994
MASE	12.59	12.152	11.793	11.43	97.987
MAE	30.445	30.005	29.583	29.176	0.272
RMSE	45.182	44.625	44.103	43.592	41.121
L1-NORM	761.12	75.013	73.956	72.94	679.99
L2-NORM	22.591	22.313	22.052	21.796	20.561
L-INF-NORM	96.392	96.392	96.128	95.776	95.307

**Table 5 sensors-23-09154-t005:** Statistical Evaluation of the Developed Deep Learning-Based Channel Estimation against Various Algorithms.

Metrics/Algorithm	AVOA	COA	GMO	EBOA	RP-WHEVO
SNR 5					
BEST	0.022696	0.023928	0.018772	0.019241	0.019241
MEAN	0.023363	0.024313	0.01904	0.019791	0.019791
WORST	0.026414	0.0262	0.020967	0.022881	0.022881
Standard Deviation	0.001231	0.000742	0.00055	0.001036	0.001036
MEDIAN	0.022696	0.023928	0.018772	0.019241	0.019241
SNR 10					
Standard Deviation	0.003382	0.004092	0.002133	0.00139	0.00139
WORST	0.039454	0.043228	0.028812	0.026738	0.026738
BEST	0.022696	0.023928	0.018772	0.019241	0.019241
MEDIAN	0.024074	0.02402	0.018237	0.019991	0.019991
MEAN	0.025101	0.025242	0.018846	0.020405	0.020405
SNR 15					
MEAN	0.040393	0.041285	0.027178	0.027986	0.027986
Standard Deviation	0.001089	0.003061	0.003061	0.001089	0.001089
MEDIAN	0.039776	0.039528	0.025421	0.027369	0.027369
WORST	0.0431	0.047179	0.033072	0.030693	0.030693
BEST	0.039776	0.039528	0.025421	0.027369	0.027369
SNR 20					
Standard Deviation	0.00158	0.002406	0.002249	0.001379	0.001379
WORST	0.032975	0.039354	0.032428	0.027749	0.027749
MEDIAN	0.024155	0.02755	0.022416	0.020721	0.020721
MEAN	0.024973	0.029315	0.024137	0.021495	0.021495
BEST	0.024155	0.02755	0.022416	0.020721	0.020721
SNR 25					
BEST	0.021298	0.021358	0.016389	0.018029	0.018029
MEDIAN	0.021298	0.021358	0.016389	0.018029	0.018029
MEAN	0.021833	0.022071	0.016992	0.018454	0.018454
WORST	0.03049	0.029783	0.024584	0.026991	0.026991
Standard Deviation	0.001467	0.001796	0.001627	0.001323	0.001323

**Table 6 sensors-23-09154-t006:** Computation complexity analysis of the implemented channel estimation scheme against various Algorithms.

Proposed Model	Computation Complexity
RP-WHEVO	$O[MaxIter*N_{pop}+1]$

**Table 7 sensors-23-09154-t007:** Validation of the Performance Offered by the Proposed Channel Estimation Model Against Recent Approaches.

Performance Measures	J-HBF-DLLPA	PSS-PARAFAC	OE-HHO	RP-WHEVO-HSCN
MEP	28.656	28.565	29.745	27.20
SMAPE	0.41345	0.40576	0.60989	39.994
MASE	0.27905	0.27474	0.28635	97.987
MAE	0.42201	0.41657	0.43106	27.22
RMSE	697.63	686.85	715.86	41.121
L1-NORM	21.101	20.828	21.553	67.999
L2-NORM	0.94369	0.94369	0.94838	20.561
L-INF-NORM	28.656	28.565	29.745	95.307

**Table 8 sensors-23-09154-t008:** Analysis of the Performance of the Offered Channel Estimation Model Regarding Diverse SNR Rates Against Various Algorithms.

Performance Measures	SNR-5	SNR-10	SNR-15	SNR-20	SNR-25
COA	0.040793	0.046796	0.056003	0.067714	0.075696
AVOA	0.048505	0.05445	0.056944	0.073091	0.077741
GMO	0.03517	0.043669	0.044523	0.046719	0.050763
EBOA	0.00513	0.01252	0.020719	0.025459	0.04144
RP-WHEVO-HSCN	0.076509	0.077266	0.081016	0.0889	0.093545

## Data Availability

Data are contained within the article.

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
