# Peer review of "A Novel Channel Estimation Framework in MIMO Using Serial Cascaded Multiscale Autoencoder and Attention LSTM with Hybrid Heuristic Algorithm"

_sensors, 2023, doi:10.3390/s23229154_

Round 1

Reviewer 1 Report

Comments and Suggestions for Authors

This paper proposed a revolutionary channel estimation methodology integrated serial cascaded multiscale autoencoder (SCMAE) and attention Long Short-Term Memory (LSTM) network by a Hybrid Heuristic algorithm, which is an interest topic. However, the reviewer has some major concerns as follow:

1) The author should highlight the main contributions compared with the prior works in the related field and state that why combined those many techniques.

2) For the research scenario, only MIMO technique is considered. However, in 5G era, the massive MIMO is widely employed. In this way, the authors need to extend the current study to the massive MIMO case and add the simulation or additional step in the proposed algorithm.

3) For the proposed RP-WHEVO algorithm, please further add the complexity analysis.

4) For the benchmark, please illustrate the reason to choose. Does this fairness is considered in comparation? E.g., with the same scenario and system parameters? 

Comments on the Quality of English Language

Moderate editing of English language required

Author Response

Comments and suggestions for Authors

This paper Proposed a revolutionary channel estimation methodology integrated serial cascaded multiscale Autoencoder (SCMAE)and attention LSTM network by a Hybrid Heuristic algorithm, which is an interested topic..However the reviewer has some major concerns as follows

  • The author should highlight the main contributions compared with the prior works in the related filed and state that they combined those many techniques.

Reply: The highlights have been compared with the prior works at the end of section 2.1.

  • For the research Scenario, only MIMO technique is considered. However, in 5G era, the massive MIMO is widely employed. In this way, the authors need to extend the current study to the massive MIMO case and the simulation or additional step in the proposed algorithm.

Reply: Since the MIMO scenario is experimented with by varying the 12 antennas, it is similar to experimenting with a massive MIMO scenario. Further, it is required it will be done in the future work as extension.

  • For the Proposed RP-WHEVO algorithm, please further add the complexity analysis.

Reply: The computation complexity of the designed method has been added in section 6.8.

  • For the bench mark, Please illustrate the reason to choose. Does this fairness is considered in comparison? Eg..with the same scenario and system parameters?

Reply: Yes, the experimentation has been conducted with the same scenario and system parameters. The experiments have been added in section 6.

Reviewer 2 Report

Comments and Suggestions for Authors

The paper focuses on the topic of accurate channel estimation in MIMO configurations. The authors in the paper propose a revolutionary channel estimation methodology that uses a synergistic combination of deep learning and heuristic techniques to handle the challenges provided by complex and dynamic wireless environments. This is an interesting research work. However, the following issues should be addressed before acceptance for publication.

1) The experimental research and performance evaluation of the paper are insufficient, so it is suggested to supplement the experiments to make the method proposed in the paper more convincing.

2) There are still many problems in the format, layout and language expression of the paper, which leads to the accurate expression of the content of the paper, which needs to be revised throughout.

3) In the performance evaluation part, the author includes traditional methods for comparison. In fact, there are a lot of excellent scientific research works in this field recently. If possible, it is suggested to include the latest representative methods in the performance evaluation part to make the methods proposed in this paper more convincing.

Comments on the Quality of English Language

The Quality of English Language should be improved. See the Comments and Suggestions for Authors.

Author Response

The paper focuses the topic of accurate channel estimation in MIMO configurations .The authors in the paper propose a revolutionary channel estimation methodology that uses a synergistic combination of deep learning and Heuristic techniques to handle the challenges provided by complex and dynamic wireless environments.This is an interested research work .However the following issues should be addressed before acceptance for publication

  • The experimental Research and performance evaluation of the paper are in sufficient, so it is suggested to supplement the experiments to make the method proposed in the paper more convincing.

Reply: The validation of the designed approach regarding diverse SNR rate has been added in section 6.10.

  • There are still many problems in the format, layout and language expression of the paper, which leads to the accurate expression of the content of the paper, which needs to be revised throughout.

Reply: The given manuscript has been modified based on the above mentioned format.

  • In the performance evaluation part, the author includes the traditional methods for comparion,In fact there are a lot of excellent scientific research works in the fields recently.If possible ,it is suggested to include the latest representative methods in the performance evaluation part to make the methods proposed in this paper more convincing. Reply: The validation of the designed approach with recent approaches has been added in section 6.9. The simulation parameters of the recommended model have been added in table 2.
  • Equations numbering and algorithm table equations are not matched.check once.

            Reply: As per your suggestion, the equation numbers has been checked successfully.

Reviewer 3 Report

Comments and Suggestions for Authors

The authors propose a framework for MIMO, however at the point where communications standards are found, this must be updated for the massive MIMO version.

There are many works in the literature that propose LSTM for channel estimation. What exactly is new about this proposal is not clear.

Much more promising methods are already proposed in the literature in which the channel is not estimated or the estimation is not necessary (Non-coherent approach with DPSK schemes or energy squemes). The need to continue using channel estimation must be argued.

Author Response

Thank you for your valuable comments concerning our manuscript entitled “Intelligent MIMO Channel: A Novel Channel Estimation Framework in MIMO using Serial Cascaded Multiscale Autoencoder and Attention LSTM with Hybrid Heuristic algorithm” for further improvements. We carefully read the manuscript and made corrections based on the comments mentioned here.  The corrections are marked in green color in the revised manuscript.

Reviewer 4 Report

Comments and Suggestions for Authors

A good paper presenting a Neural Network-based channel efficiency estimation pattern developed to estimate the channel coefficient in the wireless network. The proposed system was developed to predict the coefficient of MIMO system at the transmitter side antenna SNR 635 feedback information and reduce the error rate.

Author Response

Thank you for reviewing our manuscript

Round 2

Reviewer 1 Report

Comments and Suggestions for Authors

The comments had been addressed. 

Comments on the Quality of English Language

Minor editing of English language required

Author Response

Thank you for your valuable comments concerning our manuscript entitled “A Novel Channel Estimation Framework in MIMO using Serial Cascaded Multiscale Autoencoder and Attention LSTM with Hybrid Heuristic algorithm” for further improvements. We carefully read the manuscript and made corrections based on the comments mentioned here.  The corrections are marked in blue color in the revised manuscript.

Comments and Suggestions for authors

  1. Thanks to the authors for trying to direct my comments. Most of them have been covered ,except the argument about the need to continue investing resources in channel estimation when there are promising techniques that avoids this consumption

Reply: Thanks for your valuable comment and also we comes under your points of view the given research work the Hybrid Serial Cascaded Network (HSCN) and Hybrid Revised Position-based Wild Horse and Energy Valley Optimizer (RP-WHEVO) algorithm has been developed for enhancing the estimation of channels in the MIMO system.  Moreover, the optimization of parameters for the HSCN using RP-WHEVO has been utilized to minimize the “Root Mean Square Error (RMSE), Bit Error Rate (BER) and Mean Square Error (MSE)” of the estimated channel. Especially, few of the advantages of the optimization of parameters have been utilized to improve the efficiency of the system and also it helps to reduce the overfitting issues. It has been helped to reduce the computation time and also it generates a way to attain a better quality results. In particular, it has been utilized to enhance the estimation of channels. Accurate channel estimation allows telecommunication networks to adapt and optimize the transmitted signals, leading to improved signal quality and reduced interference.

  1. The authors have been argued in the response letter. However, This Reviewer considers that this should also appear in the manuscript, either in the introduction or motivation, to clearly if another reader may have the same doubt. The authors must include references as indicated in the previous round to argue their position of continuing to estimate the channel in the face of promising and emerging techniques that do not requires this estimation. The authors are recommended to choose the non-coherent systems based on DPSK because they are ones that have shown the best performance among all the non-coherent schemes.

Reply: Based on your suggestion, the response of DPSK based conversation in revision 2 has been added in the 4th line of section 2.2. The references based on that response has been added in the references and also it has been cited in the revised manuscript successfully. The non-coherent systems based on DPSK scheme will be considered as the future work and also the point has been added in section 7.

  1. Must be improve the research design appropriate

Reply: Based on your suggestion, the research design of this manuscript has been improved.

  1. Must be improve the methods are adequately described.

Reply: The description of the methods has been improved and also detailed description of the proposed methodology has been added in section 4.3, 5.2 and 5.3.

  1. Conclusions supported by the results must be improved and also present the results clearly.

Reply: The findings of the deigned approach have been presented in the 7th line of section 7.

  1. Must be improve the introduction provide sufficient background and include all relevant references

For references:

For example: V.M. Baeza and A.G. Armada, ”Performance and complexity Tradeoffs of several constellations for Non Coherent Massive MIMO,”2019 22nd International Symposium on Wireless Personal Multimedia Communications (WPMC),Lisbon, Portugal, 2019, pp.1-6,doi:10.1109/ WPMC48795.20199096091.

Reply: The introduction section has been improved. Additionally, the above mentioned references have been added in the reference section and also it has been cited successfully.

Reviewer 3 Report

Comments and Suggestions for Authors

Thanks to the authors for trying to direct my comments. Most of them have been covered, except the argument about the need to continue investing resources in channel estimation when there are promising techniques that avoid this consumption.

The authors have argued in the response letter. However, this Reviewer considers that this should also appear in the manuscript, either in the introduction or motivation, to clarify if another reader may have the same doubt. The authors must include references as indicated in the previous round to argue their position of continuing to estimate the channel in the face of promising and emerging techniques that do not require this estimation. The authors are recommended to choose the non-coherent systems based on DPSK because they are the ones that have shown the best performance among all the non-coherent schemes.

For example: V. M. Baeza and A. G. Armada, "Performance and Complexity Tradeoffs of Several Constellations for Non Coherent Massive MIMO," 2019 22nd International Symposium on Wireless Personal Multimedia Communications (WPMC), Lisbon, Portugal, 2019, pp. 1-6, doi: 10.1109/WPMC48795.2019.9096091.

Author Response

Thank you for your valuable comments concerning our manuscript entitled “A Novel Channel Estimation Framework in MIMO using Serial Cascaded Multiscale Autoencoder and Attention LSTM with Hybrid Heuristic algorithm” for further improvements. We carefully read the manuscript and made corrections based on the comments mentioned here.  The corrections are marked in blue color in the revised manuscript.

Reviewer 2:

  1. The Comments had been addressed

Reply: Thanks for your valuable comment.

Comments on the Quality of English Language

  1. Minor editing of the English Language required

Reply: The whole manuscript has been proof read and also the writing of this manuscript has been improved based on your suggestion.
